# The Impact of Altitude on Tick-Borne Pathogens at Two Mountain Ranges in Central Slovakia

**DOI:** 10.3390/pathogens13070586

**Published:** 2024-07-15

**Authors:** Dana Zubriková, Lucia Blaňarová, Gabriela Hrkľová, Yaroslav Syrota, Jozef Macko, Dana Blahútová, Veronika Blažeková, Michal Stanko, Klaudia Švirlochová, Bronislava Víchová

**Affiliations:** 1Institute of Parasitology, Slovak Academy of Sciences, Hlinkova 3, 040 01 Košice, Slovakiasyrota@saske.sk (Y.S.);; 2Department of Biology and Ecology, Catholic University in Ružomberok, Hrabovská Cesta 1A, 034 01 Ružomberok, Slovakia; 3I. I. Schmalhausen Institute of Zoology NAS of Ukraine, Bogdan Khmelnytsky Street, 15, 01054 Kyiv, Ukraine; 4Department of Epizootiology, Parasitology and Protection of One Health, University of Veterinary Medicine and Pharmacy in Košice, Komenského 73, 041 81 Košice, Slovakia; 5Institute of Zoology, Slovak Academy of Sciences, Dúbravská Cesta 9, 845 06 Bratislava, Slovakia

**Keywords:** abiotic factors, altitude, soil pH, tick, tick-borne pathogens, *Anaplasma*, *Borrelia*, *Babesia*/*Theileria*, *Rickettsia*, TBEV, *Ixodes ricinus*

## Abstract

Ticks are ectoparasites of a wide range of animals and are important vectors of numerous pathogens affecting humans, livestock, and pets. This study investigates possible correlations between selected factors, altitude, soil pH, and a factor called ‘amount’ (number of ticks examined in pooled samples) on the occurrence of *I. ricinus* ticks positive for selected tick-borne microorganisms. Questing *I. ricinus* ticks were collected in 2016 and 2017 across various altitudes, at two mountain ranges in central Slovakia. Tick pools were screened for the presence of *Anaplasma phagocytophilum*, *Borrelia burgdorferi* sensu lato (*Bbsl*), *Babesia*/*Theileria* spp., *Rickettsia* spp., and tick-borne encephalitis virus (TBEV) using molecular methods. Regression analysis was employed to evaluate relationships between selected factors and the occurrence of vector-borne microorganisms. This study revealed a statistically significant influence of altitude on the occurrence of *A. phagocytophilum*; increasing altitude of the sampling site was associated with increased probability of pathogen occurrence. For *Babesia*/*Theileria* spp., neither altitude nor soil pH significantly affected pathogen occurrence. The occurrence of *Bbsl* was notably impacted by both altitude and soil pH; higher altitudes were associated with a decreased probability of pathogen presence, whereas higher soil pH increased the likelihood of pathogen occurrence. The presence of *Rickettsia* in a pooled sample was not affected by altitude and soil pH, but the ‘amount’ factor was a significant predictor, increasing the probability of pathogen detection. Neither altitude nor soil pH had a significant impact on TBEV occurrence. The regression models showed moderate goodness-of-fit levels to the data, underscoring their utility in examining the role of altitude and soil pH on pathogen occurrence. However, they explained only a small portion of the overall variance in pathogen occurrence, indicating the presence of other significant factors not covered in this study.

## 1. Introduction

*Ixodes ricinus* (L., 1758) is the most widespread tick species in Europe and transmits numerous bacterial, protozoal, and viral pathogens. From the perspective of human and animal health among the most important transmitted pathogens belong *Bbsl*, *A. phagocytophilum*, *Babesia* spp., *Rickettsia* spp. and tick-borne encephalitis virus. Each pathogen has its own transmission cycle and depends on the composition and ecology of host species and vectors. *I. ricinus* has four life stages, an egg and three active stages (larva, nymph, and adult), that must feed on a vertebrate host to transform into the subsequent life stage. Larvae and nymphs can feed on more than 300 vertebrate species [1], while adult females feed only on large and medium-sized mammals [2]. The impact of hosts on the ecology and epidemiology of various pathogens is crucial. Hosts may act as amplification hosts, which increase the proportion of ticks infected by certain pathogens, or dilution hosts, which lower the prevalence of infectious agents in ticks. The presence of hosts, their abundance, and host species composition at the site affect the circulation of a pathogen in nature [3,4].

*I. ricinus* spends a significant portion of its lifespan off the host, in the environment, either unfed, engorged, in developmental diapause, or developing to the next stage. Broad-leaved and mixed forests offer favourable conditions for the species. Nevertheless, ticks can also be found in higher amounts in coniferous forests if there is sufficient precipitation [2]. Trees can strongly modify soil’s physical structure, pH, water flow, and the contents of soil organic matter and nutrients [5]. Changes in soil properties may consequently affect the quality of living conditions for soil biota, thus resulting in changes in their abundance, biomass, activity, and community structure [6,7]. Environmental soil conditions affect the survival of tick stages [8,9]. In a recent ecological study, negative binomial regression modelling indicated that limestone-based soils were more favourable for *I. ricinus* tick abundance than sandstone-based soils, and machine learning algorithms identified soil-related variables as the best predictors of *I. ricinus* tick abundance and tick-associated pathogen abundance (*Bbsl*, *Borrelia myiamotoi*, and *A. phagocytophilum*) [10]. To conclude, the biology of *I. ricinus* is very complex, and many biotic and abiotic (environmental, landscape, and anthropogenic) factors are involved in determining the abundance of ticks and the prevalence of transmitted pathogens. The interdependence of these factors is significant, but their quantification has not been well established yet [11]. The presence, diversity, and complex interrelationships between hosts, vectors, and individual components of their microbiomes (symbiotic and pathogenic) determine the infectious burden on a particular site and the associated risk of transmission of vector-borne agents.

Based on published data indicating that *Bbsl* prevalence rates vary with altitude [12,13,14,15], we primarily focused on assessing possible correlations between altitude, (but also soil pH and the number of ticks examined in pools) and the occurrence of selected vector-borne pathogens (*Bbsl*, *Babesia*/*Theileria* spp., *A. phagocytophilum*, *Rickettsia* spp., and tick-borne encephalitis virus) in *I. ricinus* pooled samples collected from two distinct mountainous areas.

## 2. Materials and Methods

### 2.1. Characterization of Sampling Sites

Tick collection sites were situated in two different mountain ranges in the Western Carpathians, the Poľana Mountain (Mt.) and the Smrekovica Mt. The Poľana Mt. is the highest volcanic mountain range in Slovakia, with the peak altitude of 1458 m above sea level (a.s.l.). Due to its exceptional natural and landscape value, it was designated as a UNESCO Biosphere Reserve, with agricultural and forestry activities restrictions. The forest is mainly composed of European beech (*Fagus sylvatica* L.), along with silver fir (*Abies alba* Mill.) and sycamore (*Acer pseudoplatanus* L.). Other tree species such as ash (*Fraxinus excelsior* L.), Norway spruce (*Picea abies* (L.) H. Karst.), and wych elm (*Ulmus glabra* Huds.) are present less frequently. Adjacent forests are dominated mostly by European beech or Norway spruce [16].

The Smrekovica Mt., with an altitude of 1532 m a.s.l., is one of the highest peaks in the Veľká Fatra mountain range. The Veľká Fatra National Park has been protected since 1973. In 2002, it was designated as a natural reserve to protect the ecosystem of natural acidophilous spruce forests (vaccinio-piceetea) with characteristics of a primeval forest.

The primary forests in the Carpathians are mostly dominated by Norway spruce, particularly at higher elevations, above 1200 m a.s.l. Other species such as European silver fir, Swiss pine, rowan, and birch can also be found in these forests [17].

There are three species of wild ungulates at both mountains: red deer (*Cervus elaphus*), roe deer (*Capreolus capreolus*), and wild boar (*Sus scrofa*), with red and roe deer being the most abundant. In the Smrekovica, there are also mouflons (*Ovis musimon*) and Northern chamois (*Rupicapra rupicapra*) present. In both areas, large predators such as wolves (*Canis lupus*), Eurasian brown bears (*Ursus arctos*), and lynx (*Lynx lynx*) prey on wild ungulates. In the Veľká Fatra and the Poľana, the bear population density is estimated to be between 5 and 11 bears per 100 square kilometres [18].

### 2.2. Soil Sampling and Tick Collection

The Poľana Mt. was sampled at six altitudes (altitudinal range from 600 to 1050 m a.s.l.) and the Smrekovica Mt. at seven altitudes (altitudinal range from 680 to 1450 m a.s.l.) (Figure 1). Soil sampling was focused on surface soil whose characteristics influence the abundance and life cycle of ticks the most. At each altitude, three trench soil pits were dug up. Soil samples were taken from each horizon of the soil pit, the surface humus horizon (the Oo horizon), followed by a layer of 5–10 cm (the A horizon) and a depth of 20–25 cm (the Bv horizon). Soil samples were dried at room temperature and processed in the laboratory. Ticks were collected once a month (from March/April to October/November 2016 and 2017). Questing ticks were collected by dragging a 1-m^2^ white wool blanket over the vegetation for one hour. Ticks were stored in pools (up to 10 nymphs, or 5 adults) at −80 °C. When the few ticks were flagged, mixed pools of nymphs and adults were prepared. Mixed pools (adults and nymphs) represented 12.24% (42 pools) of all examined pools (343 pools). Pools of *I. ricinus* ticks were examined for the presence of tick-borne pathogens (TBPs) (*Bbsl*, *Babesia/Theileria* spp., *A. phagocytophilum*, *Rickettsia* spp., and TBEV).

### 2.3. Laboratory Examination of the Soil

The pH measurement was carried out using WTW multi-meter i340 (WTW GmbH, Weilheim, Germany). The standard ratio of soil to water was 1:2.5 (10 g of soil to 25 mL of distilled water). At the surface horizon of organic matter, this ratio was 2:25 (2 g of soil sample was added to a 100 mL beaker and mixed with 25 mL of distilled water). After mixing, the soil suspensions stood for 24 h. After this time, the soil pH was measured in suspension to 2 decimal places.

### 2.4. RNA Extraction and cDNA Production

Pools of ticks were homogenized by the TissueLyzer (Qiagen GmbH, Hilden, Germany). RNA was extracted using a GeneJET RNA purification kit (Thermo Fisher Scientific, Waltham, MA, USA), eluted in 60 μL of RNase-free water, and stored at −80 °C. RNA was examined for the presence of TBEV. To detect DNA microorganisms (*Bbsl*, *Babesia/Theileria* spp., *A. phagocytophilum*, and *Rickettsia* spp.) cDNA synthesis was performed from each tick pool using RevertAid Reverse Transcriptase (Thermo Fisher Scientific, Waltham, MA, USA) and stored at −20 °C.

### 2.5. Detection of Tick-Borne Pathogens in I. ricinus Ticks

The presence of *Bbsl* genospecies was examined using PCR targeting the intergenic spacer (IGS) regions of the 5S-23S ribosomal RNA genes [19]. *Babesia/Theileria* spp. was detected using the PCR protocol targeting the 18S rRNA gene [20]. For *A. phagocytophilum* detection, primers MSP2_3F and MSP2_3R targeting a 334-bp fragment of the *msp2* gene were used in conventional PCR [21]. *Rickettsia* spp. was detected by amplifying the citrate synthase gene (*gltA*) by nested PCR [22,23]. All PCR reactions were performed using the 5× HOT FIREPol^®^ Blend Master Mix Ready to Load (Solis Biodyne, Tartu, Estonia).

To determine the presence of TBEV, the samples were examined using the real-time RT-PCR method targeting the 3′ non-coding region of the TBEV genome, as described by [24]. Real-time RT-PCR was performed using the iTaq™ Universal Probes One-Step Kit (Bio-Rad, Hercules, CA, USA). The amplification was carried out in the CFX96 Touch Real-Time PCR Detection System (Bio-Rad). Positive and negative controls were tested along with the unknown samples in each run.

### 2.6. Statistical Analysis

The TBPs’ prevalence was expressed as estimated pool prevalence (EPP). EPP with 95% confidence interval was calculated with an online pool prevalence calculator (Epitool epidemiological calculator) [25] using pooled prevalence for the variable pool size and perfect tests [26]. A geographical map was generated with the QGIS software (version 3.32) (https://qgis.org/, 2023).

### 2.7. Regression Analysis

This study used logistic regression analysis to investigate the relationship between altitude and soil pH levels and the presence or absence of TBPs (*A. phagocytophilum*, *Babesia/Theileria* spp., *Bbsl*, *Rickettsia* spp., and TBEV) in pooled samples. This approach allowed the examination of the impact of these variables on the odds of pathogen occurrence. The analysis was executed using R statistical software (version 4.2.3) [27]. Functions from the ‘tidyverse’ package [28] were utilized to manipulate the data. The ‘sjPlot’ package [29] was employed to represent and visualize models. A *p*-value less than 0.05 was considered statistically significant in the analysis.

First, multicollinearity among predictors was assessed using the Variance Inflation Factor (VIF). An initial linear model was fitted with the following predictors: altitude, pH, vegetation, slope, humus quality, soil texture, number of females per pooled sample, number of males per pooled sample, number of nymphs per pooled sample, and the total number of ticks within each pooled sample (referred to as ‘amount’ in this work). The presence of *A. phagocytophilum* served as the response variable. However, it is important to note that the choice of the dependent variable is not essential during the multicollinearity assessment. This is because the VIF measures only the relationships among predictor variables, not their dependency on the response variable. Then, models with a smaller number of predictors were fitted. After each fitting, the VIFs were calculated and predictors potentially minimizing multicollinearity were selected for the next step. This way, the following predictors were selected: altitude, pH, and ‘amount’. The analysis used the core R function for linear regression, lm, and the check_collinearity function from the ‘performance’ package [30]. Further analysis incorporated only three selected predictors: the altitude at which samples were collected, the pH level of the soil at the sampling location, and the ‘amount’ factor. These predictors were scaled and centred. Each predictor’s necessity in the specific pathogen final model was assessed by incorporating it into a model and subsequent evaluation of the model using the Akaike Information Criterion (AIC). Additionally, before the analysis, the variance in pathogen detection across different years and localities was calculated for each pathogen to assess the potential of these factors to serve as mixed effects in the model.

Generalized Linear Mixed Models (GLMMs) were implemented for *A. phagocytophilum*, *Babesia/Theileria* spp., and *Bbsl*. The models were fitted using the function *glmer* from the ‘lme4’ package [31]. The response variable was the presence of a pathogen (a binary outcome), while the predictors included the altitude and soil pH level at the collection site. Year and locality were included as random effects for the accounting of variation across different years and localities. The link function was specified as ‘logit’ to cater to the binary nature of the response variable.

Due to the low variance in detections of *Rickettsia* spp. and TBEV across different years and localities, Generalized Linear Models (GLMs) were used. These models were fitted using the function *glm* from the R environment. The presence of *Rickettsia* spp. and TBEV was used as the binary response variable in these models. The model for *Rickettsia* spp. included altitude, soil pH, and the ‘amount’ (the number of ticks in each pooled sample) as predictors, while the model for TBEV incorporated only altitude and soil pH. As with the previous models, the ‘logit’ link function was used, given the binary nature of the response variable.

The performance of each model was evaluated by determining the pseudo-R-squared values and calculating the Area Under the Curve (AUC). Marginal and conditional pseudo-R-squared values for each GLMM were computed using the *r.squaredGLMM* function from the ‘MuMIn’ package [32], while McFadden’s pseudo-R-squared for the GLMs was computed through the function *pR2* from the ‘pscl’ package [33]. The Area Under the Curve (AUC) was estimated using the function *roc* from the ‘pROC’ package version 1.18.0 [34].

## 3. Results

### 3.1. Tick-Borne Pathogen Infection in Questing I. ricinus Ticks

In total 1627 nymphs and 681 adults in 343 pools were examined for selected tick-borne pathogens at the Poľana Mt. and the Smrekovica Mt. The EPP and 95% intervals of each pathogen detected in this study at various altitudes are shown in Table 1 and Table 2. The most prevalent pathogen at both mountains was *Rickettsia* spp.: EPP = 13.07% (95%CI = 11.05–15.30) at the Poľana Mt. and EPP = 16.19% (95%CI = 12.65–20.31) at the Smrekovica Mt. *Rickettsia* spp.-positive ticks were found at all altitudes at the Poľana Mt. and up to 1370 m a.s.l. at the Smrekovica Mt. The overall EPP of *Bbsl* at the Poľana Mt. was 4.0% (95%CI = 3.06–5.11) and 3.26% (95%CI = 2.04–4.87) at the Smrekovica Mt. *Babesia/Theileria* spp. was quite prevalent with an EPP value reaching 2.98% (95%CI = 2.19–3.94) at the Poľana Mt. and 0.60% (95%CI = 0.19–1.38) at the Smrekovica Mt. The overall EPP of *A. phagocytophilum* at the Poľana Mt. was 2.15% (95%CI = 1.50–2.96) and 0.60% (95%CI = 0.19–1.38) at the Smrekovica Mt. At the Poľana Mt., *Bbsl*, *Babesia/Theileria* spp., and *A. phagocytophilum*-positive ticks were found at all altitudes. In contrast, at the Smrekovica, these pathogens were found up to 990 m a.s.l.

The overall EPP of TBEV was 0.56% (95%CI = 0.27–1.01) and 0.6% (95%CI = 0.19–1.39) at the Poľana Mt. and the Smrekovica Mt., respectively. TBEV-positive ticks were detected at altitudes of up to 900 m a.s.l. at the Poľana Mt. and up to 990 at the Smrekovica Mt.

### 3.2. Impact of Altitude and Soil pH on the Occurrence of Tick-Borne Pathogens

The logistic regression analysis results are shown in Table 3 and Table 4 and are graphically represented in Figure 2. The analysis revealed a significant influence of altitude on the occurrence of *A. phagocytophilum* in pooled samples. Increasing sampling site altitude was associated with an increased probability of pathogen presence. Conversely, soil pH had an insignificant effect on the presence of *A. phagocytophilum*. For *Babesia/Theileria* spp. occurrence, neither altitude nor soil pH significantly affected pathogen infection occurrence. In contrast to *Babesia/Theileria* spp., the occurrence of *Bbsl* was significantly influenced by altitude and soil pH. Specifically, higher altitudes were associated with lower odds of pathogen presence, while higher soil pH increased the probability of *Bbsl* occurrence. The conditional pseudo-R-squared values indicated that the predictors explained only a small part of the variance.

In the case of *Rickettsia* spp., neither altitude nor soil pH significantly affected the occurrence of the pathogen. However, the ‘amount’ was a significant predictor, associated with a higher probability of detecting the pathogen in the sample. Notably, the *Rickettsia* spp. model was the sole model where including the ‘amount’ factor improved the AIC. In the examination of TBEV detection, neither altitude nor soil pH demonstrated a statistically significant effect. The McFadden pseudo-R-squared values indicated that the *Rickettsia* and TBEV models captured only a small proportion of the variance in their dependent variables.

## 4. Discussion

The prevalence rate of *A. phagocytophilum* ranged from 0.81% to 10.7% in ticks from the Poľana Mt. and from 0 to 1.28% in ticks from the Smrekovica Mt. at various altitudes, which is consistent with other studies from Slovakia [35,36,37] and the rest of Europe [38]. Logistic regression analysis showed that the probability of detecting *A. phagocytophilum* in pooled samples of *I. ricinus* ticks increases as altitude increases. The probable explanation for this finding is the assembling of competent hosts for the pathogen and/or their higher density at higher altitudes. In Europe, the ecology and epidemiology of *A. phagocytophilum* are very complex, with several ecotypes of pathogen circulating in co-existing enzootic transmission cycles [38,39]. Ungulates such as roe deer (*Capreolus capreolus*), red deer (*Cervus elaphus*), fallow deer (*Dama dama*), and wild boar (*Sus scrofa*) are considered to be competent hosts for *A. phagocytophilum*, especially for groEL ecotype 1 and 2. Several studies from Slovakia and other European countries report high infection rates in these animals [36,40,41,42,43,44,45,46] as well as in larvae and nymphs feeding on *A. phagocytophilum* positive cervids [44]. It has been found that the prevalence of *A. phagocytophilum* is significantly higher in *I. ricinus* ticks that are found in areas with a high abundance of wild cervids, as compared to those areas where there are no cervids [47]. In Europe, ungulates tend to stay at higher altitudes during summer and move to lower sites in winter due to forage sustainability [48]. This pattern was also observed in the Carpathian, Kremnica mountains near Poľana Mt. and Smrekovica Mt., where most of the monitored red deer stayed at altitudes higher than 800 m a.s.l. from May to October [49]. Therefore, we hypothesize that the high density of ungulates dwelling at higher altitudes might be behind the correlation between rising altitude and the increased infection rates of *A. phagocytophilum* in questing *I. ricinus* ticks. However, we cannot rule out the possibility of other animal species contributing to *A. phagocytophilum* ecology at the model sites. In a recent multifactorial study, it was found that a high silt content in the soil, as well as the presence of *Bbsl*, Canidae, Cervidae, Sciuridae, or even Leporidae, positively affected the abundance of the *A. phagocytophilum.* This pathogen was mainly observed in forests with higher soil moisture and lower aerobic respiration [10].

At different altitudes, the prevalence rates of *Bbsl* were found to range from 0 to 7.04% at Smrekovica Mt. and from 1.18 to 5.13% at Poľana Mt., which is consistent with previous prevalence studies conducted in Slovakia since *Bbsl* prevalence in questing ticks was found to vary between localities [50]. Our research also revealed that the altitude level significantly affects the probability of the *Bbsl* infection in questing ticks. As the altitude increased, there was a lower probability of detecting *Bbsl*-positive ticks in pooled samples. This finding is consistent with other studies that have found a higher prevalence of *Bbsl* at lower altitudes [12,13,14,15]. The possible explanation for these findings can be the higher density of hosts such as fallow deer, roe deer, and/or red deer in higher altitudes. Cervids are mostly incompetent for all species of *Bbsl* complex that circulate in Europe, including *Borrelia burgdorferi* sensu stricto, *B. garinii*, *B. valaisiana*, *B. bavariensis*, *B. lusitaniae*, and *B. bissetii* [51]. It was also proposed that at high deer densities, a higher proportion of immature ticks parasite on deer (incompetent hosts) instead of small mammals or birds (competent hosts), thus lowering *Bbsl* prevalence in the tick population through a dilution effect [52]. The study from Norway reported a significantly higher number of *Bbsl*-infected questing *I. ricinus* ticks from islands where no cervids, but high levels of migrating birds and rodents, were present in comparison to mainland areas where cervids were abundant [47]. A recent study in New York State investigated the impact of environmental factors on the population dynamics of *B. burgdorferi* in *Ixodes scapularis* nymphs. The results revealed that altitude and several other environmental factors played a significant role. The higher altitudes were associated with a lower prevalence of *Bbsl*-positive nymphs [15].

We found a significant correlation between rising soil pH levels (from 3.66 to 6.13) and the occurrence of *Bbsl*-positive ticks in pooled samples; a rise in soil pH was linked to a higher probability of *Bbsl* occurrence. In recent research studying how ecological factors contribute to the abundance of *I. ricinus* nymphs and associated pathogens (*Bbsl*, *Borrelia myiamotoi*, and *A. phagocytophilum*), the authors found that soil characteristics are the most important factors influencing both the abundance of nymphs and also an abundance of pathogens. The most important explanatory variables for the abundance of *I. ricinus* nymphs were soil characteristics, specifically, the presence of silts, sand percentage in soil, and soil moisture. Ticks were less abundant in soils with a high percentage of sand that are more related to sandstone substrate than on clay–limestone substrate. Soil microbial respiration was also one of the most essential biological parameters for predicting tick abundance. Interestingly, most pathogens (including *Bbsl* and *A. phagocytophilum*) were more abundant among ticks in clay–limestone sites than in sandstone sites; clay–limestone soils are characterized by a higher pH than sandstone ones [10]. We hypothesize that at sites characterized by soils with higher pH levels (meaning closer to pH neutral values), there may be a higher density of competent reservoir hosts, e.g., small mammals, and/or at sites with lower pH levels, there may be a higher density of incompetent animals, such as cervids that dilute the pathogen, resulting in a lower prevalence of *Bbsl* in questing *I. ricinus* ticks.

The total infection rates of *Babesia/Theileria* spp. in pooled samples of *I. ricinus* ticks varied from 0.6% to 2.98% at Smrekovica Mt. and Poľana Mt., respectively. However, we observed variability in infection rates among collection sites at different altitudes, ranging from 2.02% to 10.8% at Poľana Mt. and from 0% to 1.32% at Smrekovica Mt. Similar infection rates of *Babesia* spp. in questing *I. ricinus* ticks were found in other studies from Slovakia [37,53,54]. According to a recent study by [55], the global pooled estimate rate (PER) for *Babesia* spp. in questing ticks in Europe is around 1.9%, with most studies reporting infection rates up to 2–3%. However, some studies have recorded higher infection rates, such as 9.7% in Germany [56]. The presence and uneven distribution of competent reservoir species in nature can explain variations in infection rates of the pathogen in questing ticks. Interestingly, no correlation was found between altitude, soil pH, or ‘amount’ factors and the presence of *Babesia*/*Theileria* spp. This finding suggests that other ecological variables affect the *Babesia*/*Theileria* spp. occurrence in *I. ricinus*.

The members of the genus *Rickettsia* are maternally inherited or transstadially transmitted symbionts of ticks and are dominant members of *I. ricinus* microbiomes [57,58,59]. The *I. ricinus* tick is a known vector for pathogenic *Rickettsia helvetica* and *R. monacensis* [60]. *Rickettsia* spp. was found to be the most common pathogen, with overall infection rates reaching 13.07% at Poľana Mt. and 16.19% at Smrekovica Mt. *Rickettsia* spp. was detected across all altitudes at Poľana Mt. and up to 1370 m a.s.l. at Smrekovica Mt. In contrast, the detection limit for all other tested pathogens was 1066 m a.s.l. at Smrekovica Mt. In Slovakia, these bacteria were detected in questing ticks with prevalences ranging from 5% to 9% [61,62,63,64]. Additionally, *I. ricinus* ticks collected from birds [63,65], small mammals [62], dogs [66], and ungulates have been found to carry DNA of *Rickettsia* spp. [41]. Our study found that neither altitude nor soil pH significantly impacted the occurrence of *Rickettsia* spp. However, the ‘amount’ was an important factor, significantly improved the model’s performance for *Rickettsia*, and increased the likelihood of the pathogen occurrence. We believe this is due to the biology of the genus *Rickettsia*, which includes maternal inheritance and transstadial transmission.

The overall prevalence of TBEV in questing *I. ricinus* ticks was found to be low at both Smrekovica and Poľana mountains. This finding is consistent with similar studies conducted in Europe and Slovakia [67,68,69,70,71]. The highest altitude with TBEV-positive ticks was 990 m a.s.l. at Smrekovica Mt. and 900 m a.s.l. at Poľana Mt. However, neither altitude nor the soil pH had a statistically significant effect on the presence of TBEV in questing ticks. TBEV can be found in small, localized microfoci [72,73]. These microfoci’s borders are poorly understood [74], making them difficult to identify. For TBEV circulation, it was suggested that the tick burden on small mammals (competent animals) will be maximal at specific deer densities and that above certain deer densities, a dilution effect will occur [75]. It is possible that the low TBEV prevalence rates identified in this study did not allow for finding a correlation with selected variables or that the selection of other ecological variables would be more appropriate for TBEV modelling.

## 5. Conclusions

This study analyzed the influence of altitude, pH of the soil, and a factor called ‘amount’ (number of ticks examined in pools) on selected tick-borne pathogens. It was found that altitude had significantly influenced the occurrence of *A. phagocytophilum*, with higher altitudes resulting in higher detection rates. However, soil pH did not affect the detection of these bacteria. In the case of *Babesia*/*Theileria*, neither altitude nor the soil pH had a significant impact on their occurrence in pooled samples of ticks. The occurrence of *Bbsl*-positive ticks in pooled samples was affected by both factors: higher altitudes resulted in lower detection rates, and at the same time, increased soil pH raised the probability of pathogen occurrence. For *Rickettsia* spp. and TBEV, a low variance was observed across different years and localities, which required a different approach. For *Rickettsia* spp., the significant predictor was the number of ticks in pools, while neither altitude nor pH had any notable effect. Similarly, altitude and soil pH did not significantly impact the occurrence of TBEV. Although all models showed moderate goodness-of-fit levels, indicating their usefulness in predicting outcomes and confirming their utility in examining the role of altitude on the probability of pathogen occurrence, they only explain a small part of the overall variance. The results suggest that other biotic and abiotic factors play a significant role in pathogen occurrence in ticks. Elucidating these factors could be a part of future research directions.

## Figures and Tables

**Figure 1 pathogens-13-00586-f001:**
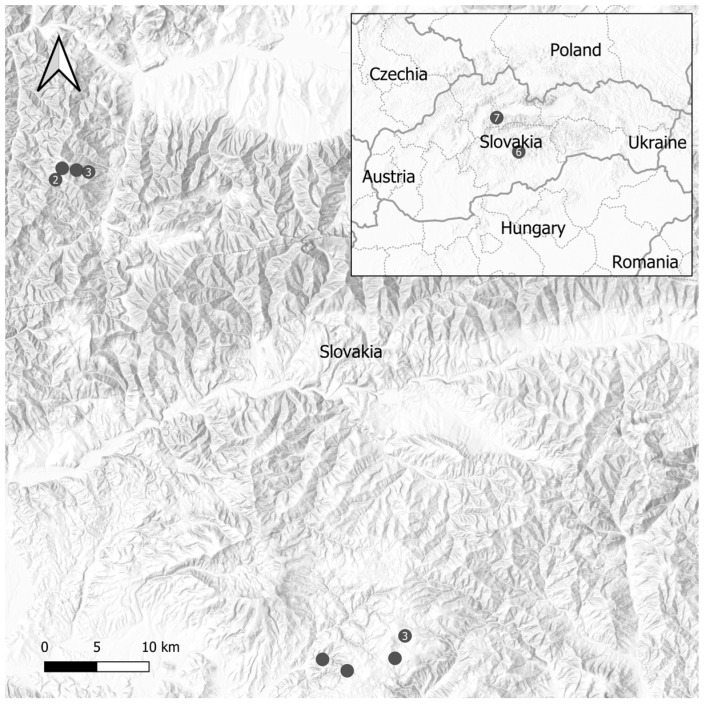
Geographic layout of tick sampling sites in the Slovak Republic. This figure illustrates the spatial arrangement of sampling sites throughout the country. Solid dots symbolize distinct sampling sites, while numbered dots depict groups of close sites bundled for enhanced visual clarity. The numerical value within these markers reflects each group’s aggregate count of sites.

**Figure 2 pathogens-13-00586-f002:**
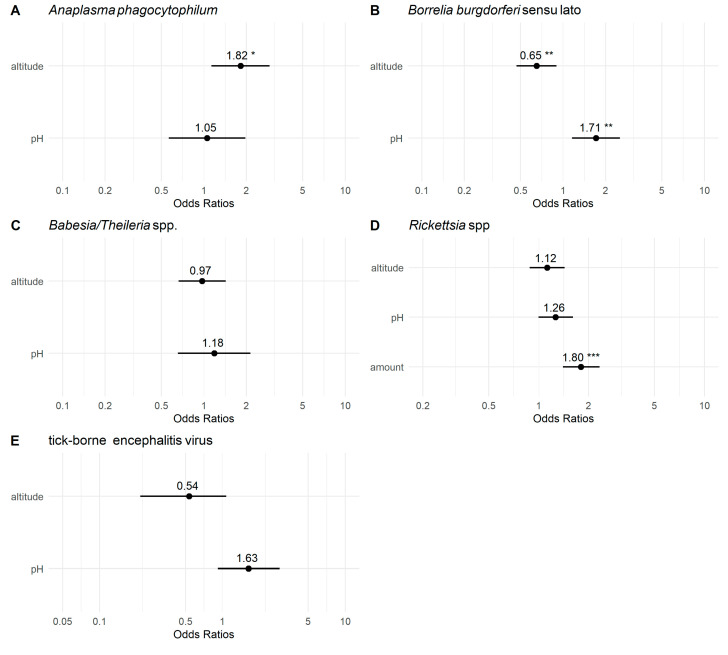
Visualisation of predictors’ effect sizes of five logistic regression models. This plot uses solid points to denote the odds ratios for each predictor. The number near the dots shows the accurate value of the odds ratio. The horizontal lines depict the 95% confidence intervals for the odds ratios. Asterisks indicate the significance of *p*-values for each predictor: * *p* < 0.05, ** *p* < 0.01, and *** *p* < 0.001.

**Table 1 pathogens-13-00586-t001:** Estimate pooled prevalence (EPP) of *Bbsl*, *Babesia*/*Theileria* spp., *A. phagocytophilum*, *Rickettsia* spp., and TBEV at Poľana Mt. in *I. ricinus* ticks.

		**No. of Ticks, Pools**				** *Bbsl* **		** *Ba.* ** **/*Th.* spp.**		** *A. phag.* **		** *Rick. * ** **spp.**		**TBEV**	
**Altitude**	**Soil pH**	**A**	**N**	**Total**	**Pools**	**+**	**EPP [CI95%]**	**+**	**EPP [CI95%]**	**+**	**EPP [CI95%]**	**+**	**MIR**	**+**	**MIR**
**600**	4.45	146	364	510	68	22	5.13 [3.29–7.51]	14	3.07 [1.74–4.93]	4	0.81 [0.25–1.86]	38	10.57 [7.6–14.18]	3	0.6 [0.15–1.55]
**700**	4.68	50	270	320	39	13	3.66 [1.85–6.36]	10	3.59 [1.81–6.23]	3	0.97 [0.24–2.5]	29	16.7 [11.27–23.69]	4	1.32 [0.41–3.03]
**800**	4.63	76	116	192	29	8	4.74 [2.19–8.67]	2	10.8 [0.18–3.28]	2	10.7 [0.18–3.27]	16	11.74 [6.96–18.17]	1	0.52 [0.0003–2.29]
**900**	4.04	96	166	262	38	3	1.18 [0.3–3.04]	5	2.02 [0.73–4.3]	8	3.38 [1.56–6.22]	19	10.27 [6.37–15.43]	1	0.39 [0.0002–1.69]
**1000**	4.07	49	60	109	21	4	4.25 [1.34–9.67]	3	2.9 [0.73–7.36]	5	5.17 [1.88–10.83]	12	17.06 [9.28–27.91]	0	0
**1050**	4.02	93	143	236	38	7	3.31 [1.43–6.33]	10	4.91 [2.48–8.48]	11	5.29 [2.77–8.91]	26	18.45 [12.31–26.23]	0	0
**Total**		**510**	**1119**	**1629**	**233**	**57**	**4.0** **[3.06–5.11]**	**44**	**2.98** **[2.19–3.94]**	**33**	**2.15** **[1.50–2.96]**	**140**	**13.07** **[11.05–15.30]**	**9**	**0.56** **[0.27–1.01]**

Note: A: adults, N: nymphs, +: positive pools, *Bbsl*: *Borrelia burgdorferi* sensu lato, *Ba.*/*Th.* spp.: *Babesia*/*Theileria* spp., *A. phag.*: *Anaplasma phagocytophilum*, *Rick.* spp.: *Rickettsia* spp., TBEV: tick-borne encephalitis virus.

**Table 2 pathogens-13-00586-t002:** Estimate pooled prevalence (EPP) of *Bbsl*, *Babesia*/*Theileria* spp., *A. phagocytophilum*, *Rickettsia* spp., and TBEV at Smrekovica Mt. in *I. ricinus* ticks.

		**No. of Ticks, Pooles**				** *Bbsl* **		** *Ba.* ** **/*Th.* spp.**		** *A. phag.* **		** *Rick. * ** **spp.**		**TBEV**	
**Altitude**	**Soil pH**	**A**	**N**	**Total**	**Pools**	**+**	**EPP** **[CI95%]**	**+**	**EPP** **[CI95%]**	**+**	**EPP** **[CI95%]**	**+**	**EPP** **[CI95%]**	**+**	**EPP** **[CI95%]**
**680**	4.46	37	268	305	39	7	2.5 [1.08–4.79]	1	0.33 [0.0002–1.44]	1	0.33 [0.0002–1.44]	24	12.9 [8.41–18.74]	1	0.33 [0.0002–1.44]
**830**	6.13	51	109	160	24	4	2.71 [0.85–6.2]	1	0.64 [0.0004–2.8]	2	1.28 [0.21–3.91]	19	26.59 [15.99–41.03]	2	1.3 [0.22–3.96]
**990**	5.89	51	106	157	25	9	7.04 [3.43–12.46]	2	1.32 [0.22–4.02]	1	0.65 [0.0004–2.82]	16	15.7 [9.3–24.3]	1	0.66 [0.0004–2.89]
**1066**	4.35	13	13	26	9	0	0	0	0	0	0	4	18.7 [6.15–38.89]	0	0
**1280**	3.66	9	8	17	8	0	0	0	0	0	0	3	18.87 [5.05–42.14]	0	0
**1370**	3.72	5	2	7	3	0	0	0	0	0	0	1	17.02 [1.05–57.62]	0	0
**1450**	5.56	5	2	7	2	0	0	0	0	0	0	0	0	0	0
**Total**		**171**	**508**	**679**	**110**	**20**	**3.26** **[2.04–4.87]**	**4**	**0.60** **[0.19–1.38]**	**4**	**0.60** **[0.19–1.38]**	**67**	**16.19** **[12.65–20.31]**	**4**	**0.60** **[0.19–1.39]**

Note: A: adults, N: nymphs, +: positive pools, *Bbsl*: *Borrelia burgdorferi* sensu lato, *Ba.*/*Th.* spp.: *Babesia*/*Theileria* spp., *A. phag.*: *Anaplasma phagocytophilum*, *Rick.* spp.: *Rickettsia* spp., TBEV: tick-borne encephalitis virus.

**Table 3 pathogens-13-00586-t003:** Generalized linear mixed models (GLMMs) for assessing the relationship between *A. phagocytophilum*, *Babesia/Theileria* spp., and *Borrelia burgdorferi* sensu lato detection in pooled samples and predictive factors: key parameters and performance estimators. Note: S_e_ denotes the standard error, σ^2^ is the mean random effects variance of the model, τ^00^ is the random intercept variance, ICC is the interclass correlation coefficient, AUC is the area under the curve, R^2^_m_ is the marginal pseudo-R-squared, and R^2^_c_ is the conditional pseudo-R-squared.

	** *A. phag.* **			** *Ba./Th. * ** **spp.**			** *Bb* ** **sl**		
**Fixed effects**									
*predictors*	*estimate*	*S_e_*	*p-value*	*estimate*	*S_e_*	*p-value*	*estimate*	*S_e_*	*p-value*
(Intercept)	−2.72	0.74	**<0.001**	−2.56	0.92	**0.006**	−2.12	1.02	**0.038**
altitude	0.60	0.24	**0.014**	−0.03	0.20	0.880	−0.43	0.17	**0.009**
pH	0.05	0.32	0.872	0.17	0.30	0.580	0.54	0.20	**0.007**
**Random Effects**									
σ^2^	3.29			3.29			3.29		
τ_00_	0.04 _year_			0.37 _year_			1.54 _year_		
	0.85 _locality_			1.15 _locality_			0.42 _locality_		
ICC _adjusted_	0.21			0.32			0.37		
N	2 _years_			2 _years_			2 _years_		
	2 _localities_			2 _localities_			2 _localities_		
**Performance**									
AUC	0.73			0.71			0.77		
R^2^_m/_R^2^_c_	0.07/0.27			0.006/0.32			0.09/0.43		
Observations	343			343			343		

Note: *Bbsl*: *Borrelia burgdorferi* sensu lato, *Ba./Th.* spp.: *Babesia/Theileria spp.*, *A. phag.: Anaplasma phagocytophilum*.

**Table 4 pathogens-13-00586-t004:** Generalized linear models (GLMs) for assessing the relationship between *Rickettsia* spp. and TBEV detection in pooled samples and predictive factors: key parameters and performance estimators. Note: S_e_ denotes the standard error, AUC is the area under the curve, and R2 is McFadden’s pseudo-R-squared.

	** *Rick. * ** **spp.**			**TBEV**		
**Fixed effect**						
*predictors*	*estimate*	*S_e_*	*p-value*	*estimate*	*S_e_*	*p-value*
(Intercept)	0.46	0.12	**<0.001**	−3.49	0.37	**<0.001**
altitude	0.12	0.12	0.346	−0.62	0.40	0.118
pH	0.23	0.12	0.056	0.49	0.29	0.085
amount	0.59	0.13	**<0.001**			
**Performance**						
AUC	0.67			0.66		
R^2^	0.06			0.05		
Observations	343			343		

Note: *Rick.* spp.: *Rickettsia* spp., TBEV: tick-borne encephalitis virus.

## Data Availability

The original contributions presented in the study are included in the article, further inquiries can be directed to the corresponding authors.

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
