# Peer review of "The Impact of Altitude on Tick-Borne Pathogens at Two Mountain Ranges in Central Slovakia"

_pathogens, 2024, doi:10.3390/pathogens13070586_

Round 1

Reviewer 1 Report

Comments and Suggestions for Authors

The authors sampled ticks from two high-altitude regions and tested and used molecular methods to screen the presence of some vector-borne pathogens. they analysed if certain biotic or abiotic factors could affect the prevalence of these pathogens in the two regions. overall this is important because it can help us in diagnosis. The methods were clearly explained.

these are a few things the authors need to improve upon

in line 21 of the abstract the word pooles, i think is a grammatical error and should be pools instead.

the introduction needs a little work, what was the basis or criteria for testing the pathogens that were selected? this should be addressed in a line or two.

also, the authors did not address why the a need to test for pathogens in those altitudes. I believe you should give the authors the importance of your research.

All the tables in the manuscript need to be redone. I did not understand what was going on in the table. your tables must be clear and not crowded if whatever you want to show does not fit in a table. find a different way to represent your data. 

In your discussions, you made mention of other Biotic and Abiotic factors that could also be looked at in the future but you did not state any of those you think should be investigated to make this work complete or make certain inferences. this should be addressed.

I don't know if it's a problem with editing, your references are very scattered and need to be redone, for clarity.

Comments on the Quality of English Language

The English is good with a few minor corrections

Author Response

Reviewer 1:

The authors sampled ticks from two high-altitude regions and tested and used molecular methods to screen the presence of some vector-borne pathogens. they analysed if certain biotic or abiotic factors could affect the prevalence of these pathogens in the two regions. overall this is important because it can help us in diagnosis. The methods were clearly explained.

these are a few things the authors need to improve upon:

Comment 1: in line 21 of the abstract the word pooles, i think is a grammatical error and should be pools instead.

Response 1: We appreciate the effort of the reviewer during manuscript reviewing. We are thankful that due to his/her valuable comments the manuscript could be improved. We corrected it and instead of “pooles” we wrote “pools”.

Comment 2: the introduction needs a little work, what was the basis or criteria for testing the pathogens that were selected? this should be addressed in a line or two.

Response 2: Thank you for this comment, we implemented those sentences in introduction section: “From the perspective of human and animal health among the most important transmitted pathogens belong Bbsl, A. phagocytophilum, Babesia spp., Rickettsia spp. and tick-borne encephalitis virus. Each pathogen has its own transmission cycle and depends on the composition and ecology of host species and vectors.”

Comment 3: also, the authors did not address why there is a need to test for pathogens in those altitudes. I believe you should give the authors the importance of your research.

Response 3: Thank you for this comment. To explain it by implementation of the paragraph into the Introduction section “Based on published data indicating that Bbsl prevalence rates vary with altitude [12–15], we primarily focused on assessing possible correlations between altitude, (but also soil pH, and the number of ticks examined in pools) and the occurrence of selected vector-borne pathogens (Bbsl, Babesia/Theileria spp., A. phagocytophilum, Rickettsia spp., tick-borne encephalitis virus) in I. ricinus pooled samples collected from two distinct mountainous areas.”

Comment 4: All the tables in the manuscript need to be redone. I did not understand what was going on in the table. your tables must be clear and not crowded if whatever you want to show does not fit in a table. find a different way to represent your data. 

Response 4: Thank you for the comment. In tables 1 and 2 we shortened the names of pathogens and therefore tables are more narrow. We made a few other changes that resulted in smaller tables, and marked them with red colour. At the moment the tables are already as simple as possible. Tables are also displayed at the moment like that because Pathogen journal wants to place tables directly in the text of the manuscript. In reality, the tables will look much better, when they will be displayed in Pathogens journal.

Comment 5: In your discussions, you made mention of other Biotic and Abiotic factors that could also be looked at in the future but you did not state any of those you think should be investigated to make this work complete or make certain inferences. this should be addressed.

Response 5: Thank you for this comment. From our point of view, the composition, and density of hosts (competent and also incompetent) is crucial for pathogen transmission. However, this factor it is quite difficult to measure, and also evaluate. Another very intriguing factor is the composition of microbiota in vectors, and subsequent microbiota interactions. Also it seems that soil characteristics can be crutial for pathogen circulation. We did not specify which abiotic or biotic factor should be studied since there are so many factors that can influence the epidemiology of vector borne diseases, and researchers can choose and make their own strategy, what to study.

Comment 6: I don't know if it's a problem with editing, your references are very scattered and need to be redone, for clarity.

Response 6: Thank you very much for this comment, we corrected few references, and we entered also one new reference: "Lüdecke, D.; Ben-Shachar, M.S.; Patil, I.; Waggoner, P.; Makowski, D. Performance: An R Package for Assessment, Comparison and Testing of Statistical Models. J. Open Source Softw. 2021, 6, 3139, doi:10.21105/joss.03139." Changes in Referencies are marked, due to revision function of the world.

Reviewer 2 Report

Comments and Suggestions for Authors

1. In supplementary data table “Raw data”, Authors indicate some additional datasets that were collected for each tick sampling site, including vegetation type, landscape (“slope”), soil texture, humus quality and sex-stage structure of tick population. I do not understand why Authors did not analyze these variables as well, though mentioning the need to do this in the Conclusions section? Please include these variables in the analysis; this will improve the significance and scientific soundness of the paper.

 2. Please check what microorganisms are studied in the paper.

Line 146 – In the text and tables, it is stated that the prevalence of Borrelia spp. is studied suggesting that combined prevalence for B. burgdorferi sensu lato and B. miyamotoi is present in the paper. However, in Methods section it is clarified that rrfA-rrlB spacer is used as a target for PCR. As far as only Borrelia burgdorferi sensu lato can be detected by this method, Borrelia spp. should be replaced by B. burgdorferi s.l. in the text and tables.

Line 148. - In the text and tables, it is stated that the prevalence of Anaplasma spp. is studied, however, in the Methods section it is stated that only one Anaplasma species, i.e. A. phaocytophilum, was detected in ticks. Please replace Anaplasma spp. with A. phagocytophilum.

 3. Please carefully check the manuscript for integrity and scientific sense. For example:

Lines 207-209 – the sense of this statement is not clear: “Interventionary studies involving animals or humans, and other studies that require 207 ethical approval, must list the authority that provided approval and the corresponding 208 ethical approval code.”

 Lines 260-262 – this paragraph makes no sense at all (“This section may be divided by subheadings. It should provide a concise and precise description of the experimental results, their interpretation, as well as the experimental conclusions that can be drawn.”).

 4. Figure 2 – There are three types of asterisks in the panels (*, ** and ***), however it is not explained in the legend which type means what. Please, clarify the designations.

 5. Please, use correct binomial species names at first mention. E.g., in the line 41, there should be “Ixodes ricinus (L., 1758)” instead of “Ixodes ricinus”. 

Author Response

Reviewer 2:

Comment 1. In supplementary data table “Raw data”, Authors indicate some additional datasets that were collected for each tick sampling site, including vegetation type, landscape (“slope”), soil texture, humus quality and sex-stage structure of tick population. I do not understand why Authors did not analyze these variables as well, though mentioning the need to do this in the Conclusions section? Please include these variables in the analysis; this will improve the significance and scientific soundness of the paper.

Response 1: We appreciate the effort of the reviewer during manuscript reviewing. We are thankful that due to his/her valuable comments the manuscript can be improved.

It is a pertinent question, and we are grateful that the Reviewer found this weakness in the manuscript. We missed some explanations in the method section. We appreciate the opportunity to clarify the methodology and rationale behind selecting predictors in detail.

To estimate potential multicollinearity, we included a comprehensive set of potential predictors in a linear model: altitude, pH, vegetation type, soil texture, humus quality, slope, number of females, number of males, number of nymphs, and total count. During the model diagnostics, we revealed issues of multicollinearity. Therefore, we employed a stepwise approach to identify and retain predictors that minimised multicollinearity.

So, we started with the full model, which included all potential predictors. The results indicated high Variance Inflation Factor (VIF) values for several predictors, mainly vegetation, pH, and altitude, suggesting severe multicollinearity. The VIF values for the initial model are summarised below:

Low correlation:

Term

VIF

VIF 95% CI

Increased SE

Tolerance

Tolerance 95% CI

amount

1.54

[1.37, 1.78]

1.24

0.65

[0.56, 0.73]

females

1.24

[1.14, 1.43]

1.11

0.80

[0.70, 0.88]

males

1.15

[1.06, 1.34]

1.07

0.87

[0.75, 0.94]

High correlation:

Term

VIF

VIF 95% CI

Increased SE

Tolerance

Tolerance 95% CI

altitude

593.61

[491.06, 717.62]

24.36

1.68e-03

[0.00, 0.00]

pH

831.07

[687.46, 1004.72]

28.83

1.20e-03

[0.00, 0.00]

vegetation

1.06e+05

[87367.69, 1.28e+05]

325.01

9.47e-06

[0.00, 0.00]

slope

2004.02

[1657.59, 2422.90]

44.77

4.99e-04

[0.00, 0.00]

humus

55.68

[46.15, 67.23]

7.46

0.02

[0.01, 0.02]

Given the high VIF values, we removed predictors with the highest VIF values. Also, there were calculation problems with VIF for nymphs, and we could not get its value for nymphs in the model. After excluding vegetation and slope, we re-fitted the model and observed the following VIF values:

Low Correlation:

Term

VIF

VIF 95% CI

Increased SE

Tolerance

Tolerance 95% CI

altitude

1.16

[1.07, 1.37]

1.08

0.87

[0.73, 0.94]

pH

1.02

[1.00, 6.64]

1.01

0.98

[0.15, 1.00]

females

2.13

[1.84, 2.52]

1.46

0.47

[0.40, 0.54]

Moderate Correlation:

Term

VIF

VIF 95% CI

Increased SE

Tolerance

Tolerance 95% CI

amount

7.40

[6.13, 9.00]

2.72

0.14

[0.11, 0.16]

nymphs

9.65

[7.96, 11.76]

3.11

0.10

[0.09, 0.13]

Nymphs and amount terms still exhibited moderate to high VIFs.

Further refinement resulted in the final model, which only included altitude, pH, and amount. The final model demonstrated acceptable VIF values, as shown below:

Low Correlation:

Term

VIF

VIF 95% CI

Increased SE

Tolerance

Tolerance 95% CI

altitude

1.14

[1.06, 1.36]

1.07

0.87

[0.73, 0.95]

pH

1.02

[1.00, 11.80]

1.01

0.98

[0.08, 1.00]

amount

1.13

[1.05, 1.36]

1.06

0.88

[0.74, 0.95]

So, the final selection of predictors – altitude, pH, and amount – was driven by the need to mitigate multicollinearity, ensuring the robustness and interpretability of our model.

Thus, our choice of predictors was not arbitrary but a deliberate decision based on statistical diagnostics to ensure our models' integrity. It is also necessary to note that other subsets of predictors in the dataset do not strongly correlate and, therefore, allow for fitting other models with low multicollinearity. However, altitude was chosen as the primary target of our study, pH as a predictor that has shown a significant impact on ticks in other studies, and the total number of individual ticks in a pooled sample as a predictor that most comprehensively represents the amount of DNA in the sample.

Considering your feedback, we have added an explanation regarding the initial choice of predictors to the appropriate section of the manuscript (Materials and Methods). Additionally, we would like to highlight that we included unused predictors in a supplementary dataset with raw data to allow other researchers to incorporate our raw data into analyses with different primary objectives. We hope these explanations clarify our approach and address your concerns.

Comment 2. Please check what microorganisms are studied in the paper.

Line 146 – In the text and tables, it is stated that the prevalence of Borrelia spp. is studied suggesting that combined prevalence for B. burgdorferi sensu lato and B. miyamotoi is present in the paper. However, in Methods section it is clarified that rrfA-rrlB spacer is used as a target for PCR. As far as only Borrelia burgdorferi sensu lato can be detected by this method, Borrelia spp. should be replaced by B. burgdorferi s.l. in the text and tables.

Line 148. - In the text and tables, it is stated that the prevalence of Anaplasma spp. is studied, however, in the Methods section it is stated that only one Anaplasma species, i.e. A. phaocytophilum, was detected in ticks. Please replace Anaplasma spp. with A. phagocytophilum.

Response 2: Thank you very much for this comment. Yes, you are right, we detected only Borrelia burgdorferi sensu lato, not Borrelia myiamotoi. Therefore we corrected it in the text, tables and figures and we replaced Borrelia spp. with Borrelia burgdorferi sensu lato or its shortcut Bbsl. The similar problem with Anaplasma, we replaced Anaplasma spp. with A. phagocytophilum elsewhere in the manuscript.

Comment 3. Please carefully check the manuscript for integrity and scientific sense. For example:

Lines 207-209 – the sense of this statement is not clear: “Interventionary studies involving animals or humans, and other studies that require 207 ethical approval, must list the authority that provided approval and the corresponding 208 ethical approval code.”

 Lines 260-262 – this paragraph makes no sense at all (“This section may be divided by subheadings. It should provide a concise and precise description of the experimental results, their interpretation, as well as the experimental conclusions that can be drawn.”).

Response 3: Thank you very much for your comment. These sentences we deleted from the text. They were just mistakes, and it happened when we placed text to the Pathogen template.

Comment 4. Figure 2 – There are three types of asterisks in the panels (*, ** and ***), however it is not explained in the legend which type means what. Please, clarify the designations.

Response 4: Thank you for this critical comment, we explained this with sentence: "Asterisks indicate the significance of p-values for each predictor: *p < 0.05, **p < 0.01 and ***p < 0.001."

 Comment 5. Please, use correct binomial species names at first mention. E.g., in the line 41, there should be “Ixodes ricinus (L., 1758)” instead of “Ixodes ricinus”. 

Response 5: We corrected it and in the first sentence of the introduction we mentioned Ixodes ricinus (L., 1758)”.